# Correlation between the Outcome of Vitrectomy for Proliferative Diabetic Retinopathy and Erythrocyte Hematocrit Level and Platelet Function

**DOI:** 10.3390/jcm11175055

**Published:** 2022-08-28

**Authors:** Keiji Sato, Tatsuya Jujo, Reio Sekine, Naoto Uchiyama, Kota Kakehashi, Jiro Kogo

**Affiliations:** Department of Ophthalmology, St. Marianna University School of Medicine, 2-16-1 Sugao, Miyamae-ku, Kawasaki 216-8511, Japan

**Keywords:** proliferative diabetic retinopathy, diabetic macular edema, hematocrit, platelet volume index

## Abstract

We investigate-d whether biomarkers such as red blood cell hematocrit (Hct), platelet count (PLT), mean platelet volume (MPV), and platelet distribution width (PDW) are useful prognostic indicators of postoperative macular edema (ME) after vitrectomy for proliferative diabetic retinopathy (PDR). A total of 42 eyes of 42 patients with PDR who underwent vitrectomy between January 2018 and May 2020 were analyzed retrospectively. We divided them into two groups according to whether treatment was required for postoperative ME and compared the relationship between Hct, PLT, MPV, and PDW and the onset of postoperative ME. The group that received postoperative treatment (group T) comprised 11 eyes of 11 patients, and the group that did not (group N) comprised 31 eyes of 31 patients. The age (years) was 52.0 ± 3.1 in group T and 60.0 ± 11.6 in group N. When appropriate statistical analysis was performed for comparison between groups, significant differences were found in age (*p* = 0.05), insulin use (*p* = 0.03), preoperative intraocular pressure (*p* = 0.05), diastolic blood pressure (*p* = 0.03), and Hct (*p* = 0.04). Multivariate logistic regression analysis was performed, and a significant difference was found in Hct (*p* = 0.02). These results suggest that Hct might be useful as a predictor of ME after PDR surgery.

## 1. Introduction

Proliferative diabetic retinopathy (PDR) is the most severe form of diabetic retinopathy (DR) characterized by the formation of new blood vessels and causes tractional retinal detachment (RD) and neovascular glaucoma (NVG), resulting in severe visual impairment [1].

PDR may occur in approximately 10% of those with type 2 diabetes mellitus (DM) [2,3], and in up to 50% of those with type 1 DM [2,4] with ≥15-year disease history. The morbidity rates of PDR are slightly higher among patients with type 2 DM who require insulin and are lower among those who do not [2]. Another important change that can cause advanced DR is diabetic macular edema (DME). DME is caused by the destruction of the blood–retinal barrier, with blood plasma leakage from the macular small vessels. DME does not cause complete blindness but often results in severe loss of visual acuity. In a large population-based study, the rate of severe visual acuity loss over 10 years of DME was 13.9% in patients with type 2 DM who did not need insulin, 25.4% in those with type 2 DM who needed insulin, and 20.1% in those with type 1 DM [5].

In recent years, it has been reported that high red blood cell (RBC) hematocrit (Hct) levels correlate with the progression of chronic kidney disease and the risk of cardiovascular thrombosis [6,7]. It has also been reported that there is a correlation between DR and Hct in the field of ophthalmology [8]. Postoperative complications of PDR include ME, which has a significant impact on visual acuity prognosis. To the best of our knowledge, there has been no report on the correlation between postoperative ME and preoperative Hct. Meta-analyses performed by Ji et al. showed that there is a relationship between DR and platelet function [9,10]. The index of platelet function is called the platelet volume index (PVI), which includes platelet count (PLT), mean platelet volume (MPV), and platelet distribution width (PDW). PDW reflects the distribution width of platelet size, and the larger the value, the more heterogeneous the platelet volume. The PVI is deeply involved in cerebral thrombosis and myocardial infarction (MI) [11]. In particular, large platelets have higher intracellular enzyme activity than smaller ones, have enhanced adhesion and release and agglutination ability, and are actively involved in thrombus formation [12]. Many studies showed the association between cardiovascular diseases (CVDs) such as MI and PVI, but not an association with DR.

Since Hct and PVI are straightforward parameters inexpensively obtained through blood sampling, and preoperative blood sampling is essential for surgery, we hypothesized that Hct and PVI could be minimally invasive predictors of prognosis in postoperative ME after PDR. Therefore, we investigated the relationships between PDR postoperative ME and preoperative Hct and PVI and examined whether these were helpful in predicting the postoperative prognosis of PDR.

## 2. Materials and Methods

### 2.1. Patients

This retrospective observational study examined 42 eyes in 42 patients who underwent pars plana vitrectomy (PPV) for PDR at St. Marianna University School of Medicine from January 2018 to May 2020. Data were obtained by tracking patient charts. The patients were divided into two groups depending on the presence or absence of antivascular endothelial growth factor (anti-VEGF) vitreous injection and/or triamcinolone acetonide subtenon (STTA) injection for postoperative ME. The eyes were divided into the postoperative treatment (T) group and postoperative nontreatment (N) group and the correlation with preoperative Hct was compared between them. Indications for PPV included the following: recurring vitreous hemorrhage (VH) despite the use of maximal panretinal photocoagulation (PRP); dense premacular subhyaloid hemorrhage; combined tractional and rhegmatogenous RD by the fibrovascular membrane (FVM); tractional RD involving or threatening the macula. Exclusion criteria were a history of past vitreous surgery, the presence of existing NVG, and the use of intraoperative silicon oil. We also excluded cases in which both eyes were in different groups, in cases in which PPV was performed on both eyes. When both eyes were in the same group, the right eye was selected as the target eye.

### 2.2. Surgical Technique

All surgeries were performed by one skilled surgeon (J.K.). All patients underwent standard PPV under regional anesthesia. In patients with cataracts, phacoemulsification and aspiration were performed simultaneously, and an acrylic foldable intraocular lens was placed in the capsular bag. All PPVs were conducted using a 25-gauge or 27-gauge 3-port system (Alcon Constellation; Vision System, Fort Worth, TX, USA) and a high-speed vitreous cutter (10,000 cycles/min), with intravitreal injection of triamcinolone acetonide to visualize the vitreous gel and vitreoretinal adhesions. A valved cannula entry system was used. In cases with FVM, segmentation was performed after core vitrectomy, followed by delamination of FVM and removal of the posterior hyaloid face. Vitreous base shaving was performed under scleral depression, and blood clots in the peripheral vitreous skirt were also removed. Intraoperative bleeding was controlled either by endodiathermy or by increasing the reflux pressure. An endolaser was applied to complete PRP up to the ora serrata. When performing internal limiting membrane (ILM) peeling, brilliant blue G was used in all cases.

In some patients with combined traction-rhegmatogenous RD or RD caused by iatrogenic tear, fluid–air exchange was performed. In severe cases, sulfur hexafluoride (SF6) gas was injected at the end of surgery. Patients who underwent fluid–gas exchange were instructed to remain face down for 3 to 7 days. For patients taking an anticoagulant for underlying systemic disease, the anticoagulant was discontinued for 1 week preoperatively if possible and resumed within 2 weeks postoperatively.

### 2.3. Clinical Data Analysis

Preoperative examination parameters were age; gender; PLT; MPV; PDW; hemoglobin A1c (HbA1c); axial length; preoperative intraocular pressure (IOP); preoperative logarithm of the minimum angle of resolution (logMAR) best-corrected visual acuity (BCVA); Hct; hemoglobin (Hb); creatinine value (Cr); estimated glomerular filtration rate (eGFR); low-density lipoprotein cholesterol (LDL-chol); high-density lipoprotein cholesterol (HDL-chol); triglyceride (TG); blood urea nitrogen (BUN); pulse pressure; mean blood pressure; BUN/Cr ratio; systolic blood pressure; diastolic blood pressure; fasting blood glucose; body mass index (BMI); hypertension (HT); whether an oral antiplatelet, oral anticoagulant, oral hypoglycemic, oral metformin, oral statin, and oral diuretic were used; whether insulin was used; smoking habit; presence or absence of artificial dialysis; preoperative PRP and preoperative anti-VEGF vitreous injection; preoperative lens condition; whether cataract surgery and/or ILM peeling were performed; whether intraoperative gas replacement was performed and the type of gas; postoperative logMAR BCVA (at 1, 3, 6, 12 months); observation period (months). In all eyes, postoperative central subfoveal thickness (CST) was derived from the software (Cirrus 3.0; Carl Zeiss Meditec, Inc., Dublin, CA, USA) provided by the manufacturer. Postoperative ME was defined as more than 300 μm CST occurring within 12 months after surgery. We divided the two groups according to the presence or absence of postoperative ME treatment (groups T and N, respectively) and compared the results between them.

### 2.4. Statistical Analysis

Statistical processing was performed using JMP 13 (SAS Institute Inc., Cary, NC, USA). In the analysis of continuous variables, the Shapiro–Wilk test was performed to examine the normality of each study parameter. The homogeneity of variance was determined using the F-test. Based on the results, we then selected Student’s *t*-test, Welch’s *t*-test, or the Wilcoxon rank-sum test. The appropriate tests were performed for each parameter to compare significant differences between the two groups. In the analysis of nominal variables, the Pearson chi-square test and Fisher’s exact test were performed to compare significant differences between the two groups. A *p*-value of <0.05 was considered statistically significant. Multivariate logistic regression analysis was performed with metformin and high statin levels considered to be confounders of PVI. To rule out intravascular dehydration as a confounding factor, a multivariate logistic regression analysis was performed using the BUN/Cr ratio as an indicator of intravascular dehydration. In these multivariate logistic regression analyses, the presence of anti-VEGF drug vitreous and/or STTA injection within 12 months after surgery were selected as dependent variables. To evaluate the test performance of Hct and calculate the cutoff value, a univariate logistic regression analysis was performed, and the receiver operating characteristic (ROC) curve was extracted. The AUC was calculated from the ROC curve obtained, and the cutoff value was calculated using the Youden index.

## 3. Results

Table 1 shows the patients’ baseline characteristics. There were 11 eyes in 11 patients in the T group and 32 eyes in 32 patients in the N group. One patient was included in both the T and N groups in the left and right eye, respectively. Gender, presence or absence of HT, hyperlipidemia, oral hypoglycemic agent use, oral hyperlipidemia, oral anticoagulant use, oral antiplatelet agent use, preoperative logMAR BCVA, axial length, smoking history, BMI, dialysis, the presence or absence of preoperative high systolic blood pressure, fasting blood pressure, preoperative anti-VEGF vitreous injection, and preoperative PRP were compared between the two groups, and no significant differences were observed in any parameter (*p* > 0.05). However, there were significant differences in age, HT insulin use, preoperative IOP, and high diastolic blood pressure (all *p* < 0.05). Seven eyes (63.6%) in group T and eight eyes (25.8%) in group N (*p* = 0.03) were from patients using insulin. Patient age was 52.0 ± 3.1 years (mean ± SD) in group T and 60.0 ± 11.6 years (mean ± SD) in group N (*p* = 0.05). Preoperative IOP was 15.9 ± 2.9 mmHg (mean ± SD) in group T and 13.5 ± 3.5 years (mean ± SD) in group N (*p* = 0.03). Diastolic blood pressure was 69.4 ± 10.3 mmHg (mean ± SD) in group T and 78.1 ± 10.9 years (mean ± SD) in group N (*p* = 0.03).

Table 2 shows the data from patients’ baseline blood tests. The Hct was 42.0 ± 5.6 (%; mean ± SD) in group T and 37.6 ± 5.6 (%; mean ± SD) in group N (*p* = 0.04). No significant differences were observed in any of the other parameters, i.e., HbA1c, fasting blood glucose, PLT, MPV, PDW, HbA1c, LDL-chol, HDL-chol, TG, eGFR, Hb, Cr, BUN/Cr, and BUN (*p* > 0.05).

Table 3 shows patient characteristics after cataract surgery, ILM peeling, and gas tamponade by type of gas injected. The characteristics examined were postoperative logMAR BCVA, postoperative IOP, and observation period. A significant difference was observed only in postoperative IOP at 6 months (*p* = 0.03).

Table 4 shows the multivariate logistic regression analysis for risk factors associated with PVI with metformin and statins as confounders. The *p*-values were: PLT (10^3^/µL) = 0.66 (odds ratio (OR), 95% confidence interval (CI) 1.00 [0.99–1.02]; MPV (fl) = 0.09 (OR (95% CI) = 0.38 [0.13–1.16]); PDW (fl) = 0.08 (OR (95% CI) = 7.41 [0.80–68.6]); oral statin use, eyes (%) = 0.82 (OR (95% CI) = 1.20 [0.25–5.73]), and oral metformin use, eyes (%) = 0.30 (OR (95% CI) = 2.34 [0.46–11.8]).

Multivariate logistic regression analysis was performed to rule out intravascular dehydration, a confounding factor for Hct (Table 5). The BUN/Cr ratio was used as an index of intravascular dehydration and showed a *p*-value of 0.19 (OR (95% CI) = 1.08 [0.97–1.21]), and the *p*-value for Hct was 0.02 (OR (95% CI) = 0.84 [0.72–0.98]). Figure 1 shows the ROC curve obtained from the results of univariate logistic regression analysis with the objective variable of treatment for postoperative ME and the explanatory variable of Hct. The sensitivity was 72.7%, specificity 64.5%, area under the curve (AUC) 0.71, and cutoff value 39.3%.

The ROC curve was obtained from the results of univariate logistic analysis of Hct. The sensitivity was 72.7%, specificity 64.5%, AUC 0.71, and cutoff value 39.3%.

## 4. Discussion

This study investigated prognostic biomarkers for postoperative ME after PDR surgery, with the main focus on examining the usefulness of Hct and PVI as preoperative prognostic markers. Hct, but not PVI, showed a significant difference between the two groups of eyes (Table 2). Previous studies showed that metformin and statins lower MPV [13,14]. Therefore, we used metformin and statins as confounders and performed a multivariate logistic regression analysis for PVI (Table 4). However, there was no significant difference in PLT, MPV, and PDW in the eyes of patients receiving those two agents. These results suggest that higher Hct might cause postoperative ME after PDR surgery. Romero et al. also reported the relationship between Hct and ME [8]. Spaide found that Muller cells are an important factor in the etiology of ME [15]. Muller cells cross most of the thickness of the retina and are the major determinant of retinal fluid movement through the aquaporin 4 channel. The footplates of the Muller cells form the ILM of the retina, and Muller cell processes surround the retinal vessels of the superficial and deep vascular plexus. The fluid in the retina moves from the superficial vascular plexus toward the deep vascular plexus, and the excess is removed from the retina by being sent into the deep vascular plexus by the action of Muller cells. Deep vascular plexus occlusion might cause ME by blocking Muller cell function.

High Hct levels are thought to inhibit the action of nitric oxide (NO) [16,17] and correlate with the progression of chronic kidney disease and the risk of cardiovascular thrombosis [18]. NO has a vasodilatory effect dependent on the vascular endothelium and decreased NO activity may inhibit vasodilation and cause obstruction of the deep vasculature by blood cells [16]. Higher Hct could cause obstruction of the deep retinal vasculature and lead to ME by blocking Muller cell function. It was reported that higher Hct levels increase blood viscosity [19,20]. Moreover, increased blood viscosity might cause vascular occlusion. Shiga et al. showed that blood viscosity is affected by Hct concentration, the erythrocyte deformation phenomenon, and the depletion of erythrocyte aggregation [21]. A high Hct concentration increases blood viscosity by increasing collision between RBCs [22]. RBC deformation in blood vessels reduces flow resistance, although increased blood viscosity decreases the deformability of RBCs. When RBC deformability decreases, the flow resistance also increases the viscosity of blood [23]. The increase in viscosity reduces the rate of blood slippage in the blood vessels, and the low-rate area causes the assembly of RBCs, which leads to a further increase in viscosity [23,24].

As described above, the decrease in NO action and increase in blood viscosity might cause a reduction in blood flow in the deep retinal vasculature, which impairs the water excretion action of Muller cells and causes ME. In addition, it was reported that smoking [25,26,27], HT, hyperglycemia [28], DM [29], elevated HbA1c, insulin use, alcohol intake [30], total cholesterol, and LDL-chol [31] are involved in the severity of DR and arteriosclerosis. Therefore, these factors might reduce the action of NO in the blood vessels, increase blood viscosity, and increase the risk of ME.

In the present study, insulin use was significantly higher in group T than in group N (Table 1, *p* = 0.03). Several reports identified insulin use as a risk factor for DME [2,32], and our results supported that finding. Insulin use may reflect poor glycemic control, and it was reported that hyperglycemia reduces vascular endothelial function [33,34] and that DM reduces the use of NO [35]. Therefore, insulin use might be a risk factor for PDR postoperative ME. As shown in Table 1 and Table 3, high preoperative (*p* = 0.05) and postoperative IOP (*p* = 0.03) might be risk factors for PDR postoperative ME. In this study, patients with obvious preoperative NVG were excluded, but it is possible that not all patients underwent gonioscopy and that our groups included hidden NVG. In a future study, it will be necessary to perform gonioscopy on all patients at the screening stage.

Contrary to expectations, the group with high diastolic blood pressure in the present study did not require treatment (*p* = 0.03). High diastolic blood pressure is a risk factor for DR, heart disease, and CVD [35,36]. It was also reported to be a risk factor for DME [36,37]. Further studies with larger sample sizes will be necessary to determine the role of HT in PDR. Table 4 shows the results of multivariate logistic regression analysis for PVI. Metformin and statins lower MPV, suggesting that they may act as confounders of PVI. As a result of multivariate logistic regression analysis, even after removing the effects of these confounding factors, no significant difference was found in any parameter of PVI.

Table 5 shows the results of multivariate logistic regression analysis for Hct. Both the BUN/Cr ratio and Hct are used as indicators of intravascular dehydration, and it is possible that intravascular dehydration may act as a confounding factor for Hct. Therefore, confounding factors were excluded by using the BUN/Cr ratio as an index of intravascular dehydration. The results showed that Hct might be useful as a preoperative predictor of PDR postoperative ME. To investigate the performance of Hct as a predictor of the prognosis of PDR postoperative ME, a univariate logistic regression analysis was performed and an ROC curve was created (Figure 1). The ROC curve showed that the AUC was 0.71 (*p* = 0.04), and the cutoff value of Hct was 39.3%. The sensitivity based on this cutoff value was 72.7%, and the specificity was 64.5%. Although the AUC of 0.71 was somewhat low in terms of test performance, Hct might be useful as a predictor of PDR postoperative ME. Insulin use and preoperative IOP might also be useful prognostic factors.

## 5. Limitations

The sample size in this retrospective was small, and no statistical sample size calculations were conducted. However, for Hct, a sample size of 42 patients gave post hoc powers of 0.69 to detect 4.37% of the smallest difference in the mean between the two groups under the conditions of the two-sided significance level (α) of 0.05. To test post hoc powers, JMP 13 (SAS Institute Inc., Cary, NC, USA) was used.

We also examined the post hoc powers of other parameters, with univariate analysis results of *p* < 0.1. We, therefore, confirmed that the power of detection was 0.60 or more for many parameters. However, a post hoc power of 0.60 is not a sufficient value and indicates that the difference between the two groups might not have been detected correctly. When both eyes of a patient were in the same group, the right eye was selected as the target in this test, but selection bias could have occurred and affected the test results.

## 6. Conclusions

No significant difference in PVI was found between the two groups in this study. Although the AUC was 0.71, and the predictive power was relatively low (cutoff value = 39.3%), Hct might be useful as a predictor of ME after PDR surgery. The decrease in blood flow in the deep retinal plexus might inhibit the water-removing action of Muller cells in the retina and cause ME. Two possible causes of decreased blood flow in deep retinal blood vessels are vascular dilatation disorder due to reducing NO action caused by decreased vascular endothelial cell function and vascular occlusion due to increased blood viscosity. No significant results were obtained in this study on the risk factors for arteriosclerosis such as HT, high LDL-chol, low HDL-chol, hyperlipidemia, and hyperglycemia, although those factors might cause a decrease in vascular endothelial function and an increase in blood viscosity and therefore could be predictors of ME after PDR surgery.

This study was conducted on a small number of eyes, and it will be necessary to conduct further studies in the future involving more eyes from a larger patient group.

## Figures and Tables

**Figure 1 jcm-11-05055-f001:**
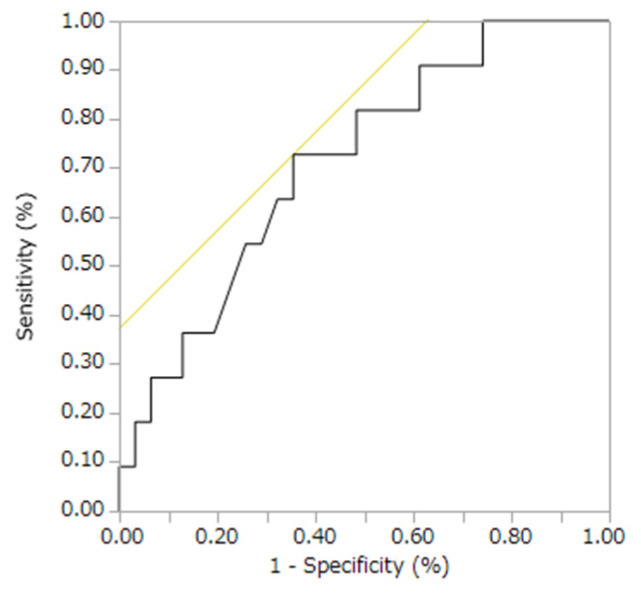
ROC curve of Hct.

**Table 1 jcm-11-05055-t001:** Patient characteristics and baseline data.

Characteristic	Group T	Group N	*p* Value
No. of patients	11	31	
No. of eyes	11	31	
Sex (male/female)	9/2	24/7	1.0
Age (years; mean ± SD)	52.0 ± 3.1	60.0 ± 11.6	0.05
Hypertension, eyes (%)	6 (54.6)	26 (83.9)	0.09
Hyperlipidemia, eyes (%)	4 (36.4)	12 (38.7)	1.0
Oral hypoglycemic, eyes (%)	6 (54.6)	24 (77.4)	0.24
Oral metformin, eyes (%)	4 (36.4)	7 (22.6)	0.44
Insulin use, eyes (%)	7 (63.6)	8 (25.8)	0.03
Oral anticoagulant, eyes (%)	1 (9.1)	5 (16.1)	1.0
Oral antiplatelet, eyes (%)	4 (36.4)	7 (22.6)	0.44
Oral statin, eyes (%)	5 (45.5)	10 (32.3)	0.48
Oral diuretic, eyes (%)	5 (45.5)	16 (51.6)	1.0
Preoperative IOP (mmHg; mean ± SD)	16.1 ± 3.1	13.6 ± 3.8	0.05
Preoperative logMAR BCVA (mean ± SD)	1.1 ± 0.7	1.1 ± 0.8	0.87
Axial length (mm; mean ± SD)	24.3 ± 1.5	24.1 ± 2.0	0.40
Lens status; phakia/pseudophakia (eyes)	8/3	22/9	1.0
Surgical purpose, eyes	11	31	0.41
VH, eyes (%)	6 (5)	23 (74.2)	
RD, eyes (%)	3 (27.3)	3 (9.7)	
DME, eyes (%)	1 (9.1)	1 (3.2)	
Fibrovascular proliferation, eyes (%)	1 (9.1)	3 (9.7)	
Neovascularization elsewhere, eyes (%)	0 (0)	1 (3.2)	
Smoking, eyes (%)	6 (54.6)	20 (64.5)	0.72
Duration of smoking (years; mean ± SD)	12.8 ± 9.6	14.6 ± 10.4	0.54
BMI (kg/m^2^; mean ± SD)	24.3 ± 2.9	27.0 ± 6.5	0.38
Systolic blood pressure (mmHg; mean ± SD)	131.6 ± 20.9	133.1 ± 17.8	0.82
Diastolic blood pressure (mmHg; mean ± SD)	69.4 ± 10.3	78.1 ± 10.9	0.03
Pulse pressure (mmHg; ± SD)	62.3 ± 20.0	55.0 ± 12.4	0.46
Mean blood pressure (mmHg; mean ± SD)	90.1 ± 10.5	96.4 ± 12.3	0.14
Preoperative anti-VEGF vitreous injection, eyes (%)	4 (36.4.2)	6 (19.4)	0.41
Previous PRP, eyes (%)	6 (54.6)	21 (67.7)	0.48
Dialysis, eyes (%)	3 (27.3)	10 (65.7)	1.0
Preoperative DME, eyes (%)	6 (54.6)	7 (22.6)	0.14

SD: standard deviation; IOP: intraocular pressure; logMAR: logarithm of the minimum angle of resolution; BCVA: best-corrected visual acuity; VH: vitreous hemorrhage; RD: retinal detachment; DME: diabetic macular edema; BMI: body mass index; VEGF: vascular endothelial growth factor; PRP: panretinal photocoagulation.

**Table 2 jcm-11-05055-t002:** Comparisons of clinical characteristics between group T and group N.

Characteristic	Group T	Group N	*p* Value
HbA1c (%; mean ± SD)	8.1 ± 2.2	6.8 ± 1.1	0.09
Fasting blood glucose (mg/dL; mean ± SD)	160.4 ± 89.3	136.4 ± 44.0	0.52
PLT (10^3^/µL; mean ± SD)	235.9 ± 36.4	237.9 ± 65.4	0.92
MPV (fl; mean ± SD)	8.5 ± 1.1	8.2 ± 0.9	0.51
PDW (fl; mean ± SD)	16.9 ± 0.5	17.1 ± 0.5	0.43
Hct (%; mean ± SD)	42.0 ± 5.6	37.6 ± 5.6	0.04
LDL (mg/dL; mean ± SD)	134.8 ± 52.7	101.1 ± 42.3	0.06
HDL (mg/dL; mean ± SD)	54.0 ± 9.3	51.1 ± 18.4	0.26
TG (mg/dL; mean ± SD)	148.9± 111.0	150.7 ± 129.4	0.89
eGFR (mL/min/1.73 m^2^; mean ± SD)	42.8 ± 21.7	46.1 ± 57.7	0.68
Hb (g/dL; mean ± SD)	13.7 ± 2.0	12.6 ± 2.2	0.12
Cr (mg/dL; mean ± SD)	2.6 ± 3.2	3.0 ± 2.7	0.51
BUN/Cr ratio (mean ± SD)	14.5 ± 6.9	16.1 ± 8.7	0.71
BUN (mg/dL; mean ± SD)	26.7 ± 22.2	31.4 ± 16.1	0.06

SD: standard deviation; HbA1c: hemoglobin A1c; PLT: platelet count; MPW: mean platelet volume; PDW: platelet distribution width; Hct: hematocrit; LDL: low-density lipoprotein cholesterol; HDL: high-density lipoprotein cholesterol; TG: triglyceride; eGFR: estimated glomerular filtration rate; Hb: hemoglobin; Cr: creatinine value; BUN: blood urea nitrogen.

**Table 3 jcm-11-05055-t003:** Postoperative clinical data.

Characteristic	Group T	Group N	*p* Value
Cataract surgery, eyes (%)	7 (63.6)	18 (58.1)	1.0
ILM peeling, eyes (%)	8 (72.7)	22 (71.0)	1.0
Gas tamponade, eyes (%)	2 (18.2)	8 (25.8)	0.84
Air, eyes (%)	1 (9.1)	3 (9.7)	
SF6, eyes (%)	1 (9.1)	5 (16.1)	
C3F8, eyes (%)	0 (0)	0 (0)	
Postoperative logMAR BCVA (mean ± SD)			
1 month	0.49 ± 0.43	0.51 ± 0.53	0.90
3 months	0.38 ± 0.35	0.36 ± 0.36	0.80
6 months	0.18 ± 0.18	0.31 ± 0.38	0.76
12 months	0.02 ± 0.07	0.18 ± 0.29	0.43
Postoperative IOP (mmHg; mean ± SD)			
1 month	15.2 ± 4.7	14.9 ± 5.1	0.78
3 months	14.3 ± 4.3	14.7 ± 4.9	0.81
6 months	17.7 ± 5.3	13.8 ± 3.7	0.03
12 months	16.4 ± 3.8	13.8 ± 3.0	0.11
Follow-up period (months; mean ± SD)	7.6 ± 4.3	9.3 ± 3.6	0.21

ILM: internal limiting membrane; SF6: sulfur hexafluoride; C3F8: octafluoropropane; logMAR: logarithm of the minimum angle of resolution; BCVA: best-corrected visual acuity; SD: standard deviation; IOP: intraocular pressure.

**Table 4 jcm-11-05055-t004:** Multivariate logistic regression analyses of risk factors associated with PVI.

Analysis (*n* = 42)	Factor	Odds Ratio (95% CI)	*p* Value
Multivariate logistic regression	PLT (10^3^/µL)	1.00 (0.99–1.02)	0.66
	MPV (fl)	0.38 (0.13–1.16)	0.09
	PDW (fl)	7.41 (0.80–68.6)	0.08
	Oral statin, eyes (%)	1.20 (0.25–5.73)	0.82
	Oral metformin, eyes (%)	2.34 (0.46–11.8)	0.30

PVI: platelet volume index; CI: confidence interval; PLT: platelet count; MPW: mean platelet volume; PDW: platelet distribution width.

**Table 5 jcm-11-05055-t005:** Multivariate logistic regression analyses of risk factors associated with Hct.

Analysis (*n* = 42)	Factor	Odds Ratio (95% CI)	*p* Value
Multivariate logistic regression	Hct (%)	0.84 (0.72–0.98)	0.02
	BUN/Cr ratio	1.08 (0.97–1.21)	0.19

Hct: hematocrit; CI: confidence interval; BUN: blood urea nitrogen; Cr: creatinine.

## Data Availability

The data presented in this study are available on request from the corresponding author. The data are not publicly available due to ethics committee permission.

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
