# Peer review of "Correlation between the Outcome of Vitrectomy for Proliferative Diabetic Retinopathy and Erythrocyte Hematocrit Level and Platelet Function"

_jcm, 2022, doi:10.3390/jcm11175055_

Round 1

Reviewer 1 Report

Dear Authors, 

I wished to submit my review for the article titled: 

"Correlation between the outcome of vitrectomy for proliferative diabetic retinopathy and erythrocyte hematocrit level and  platelet function"

The introduction is well structured, and the discussion deeply analyses the results. The article is very interesting and the authors should be commended for their work.

However, a few points should be discussed:

1. Abstract: It would be useful to reduce the statistical analysis description. It could be better to further point out the results in this section.

2. Methods: 2.4. Statistical Analysis:

a. The sample size calculation and test for normality were not mentioned. Please add this information. 

 b. The authors assessed both eyes of some patients. (PAGE 3- LINE 125) Measurements obtained from the right and left eye of a subject are often correlated, whereas many statistical tests assume observations in a sample are independent. Hence, data collected from both eyes cannot be combined without taking this correlation into account. According to Armstrong et al. (Armstrong RA. Statistical guidelines for the analysis of data obtained from one or both eyes. Ophthalmic Physiol Opt. 2013 Jan;33(1):7-14. doi: 10.1111/opo.12009. PMID: 23252852) if one eye is studied and both are eligible, then it should be chosen at random, or two-eye data can be analyzed incorporating eyes as a ‘within subjects’ factor.

How did you manage it?

Author Response

Thank you very much for your favorable reviews of our submission and helpful comments and suggestions to improve the paper. Based on our knowledge and the limited data currently at hand, we have incorporated the suggested changes in the manuscript and hope that they address the points raised sufficiently.

To Reviewer #1

  1. Abstract: It would be useful to reduce the statistical analysis description. It could be better to further point out the results in this section.

- Thank you for that suggestion. As you pointed out, I reduced the references to statistics and added more to the results.

  1. Methods: 2.4. Statistical Analysis:
  2. The sample size calculation and test for normality were not mentioned. Please add this information

 - Thank you. In the analysis of continuous variables, the Shapiro-Wilk test was performed to examine the normality of each study item, and the F-test was performed to determine the homogeneity of variance. Based on these results, we selected the Student’s t-test, Welch’s t test, or Wilcoxon rank-sum test for further analyses.

- As you pointed out, the sample size in this study was small and no statistical sample size calculations were conducted. However, the Hct sample size of 42 patients gave post hoc powers of 0.69 to detect 4.37% of the smallest difference in mean between the two groups under the condition of a two-sided significance level (α) of 0.05. To test post hoc powers, JMP 13 (SAS Institute Inc., Cary, NC, USA) was used.

We also examined the post hoc powers of other parameters, for which the results of the univariate analysis were p < 0.1. We confirmed that the power of detection was 0.6 or more for many parameters. However, as you pointed out, a power of 0.6 is not a sufficient value, and therefore we added the low power as a limitation of this study.

  1. The authors assessed both eyes of some patients. (PAGE 3- LINE 125) Measurements obtained from the right and left eye of a subject are often correlated, whereas many statistical tests assume observations in a sample are independent. Hence, data collected from both eyes cannot be combined without taking this correlation into account. According to Armstrong et al. (Armstrong RA. Statistical guidelines for the analysis of data obtained from one or both eyes. Ophthalmic Physiol Opt. 2013 Jan;33(1):7-14. doi: 10.1111/opo.12009. PMID: 23252852) if one eye is studied and both are eligible, then it should be chosen at random, or two-eye data can be analyzed incorporating eyes as a ‘within subjects’ factor.

 - Thank you for that suggestion. I have read the paper you presented and learned a lot.

The correlation between the two eyes was tested by the ICC, but it was difficult to examine the correlation between the two eyes due to the large number of parameters to be examined. Therefore, in the four cases of eight eyes in which both eyes were in the same group, the test was performed on the right eye as the target eye. In addition, the 2 eyes of 1 patient included in different groups were excluded from the test. The possibility of selection bias due to the selection of the right eye as the target was described as a limitation of this test. (page 17, 1st para, line1-12)

Reviewer 2 Report

The authors present the effect of pre-operative hematocrit levels on post-operative macular edema in proliferative diabetic retinopathy.

Although an interesting concept, the manuscript suffers from major grammatical and methodological flaws.

1. The number of eyes in the "T- group" is too small to provide any significant power to the results. Did the authors calculate the sample size? What is the power of the study?

3. Methods: 

a. Page 2 line 71: The word "invisible" needs to be rephrased. The indications of PPV need to be explained in detail. Surgery is not required for all types of fibrovascular membranes. The authors need to specify.

b. How were the criteria for diagnosis of macular edema? This needs to be explained further. Was OCT done? was it diagnosed clinically? Were the macular thickness measurements done? Were they centre-involving or non-centre involving?

2. The multivariate logistic regression tables are unclear. What are the dependent variables in these two tables?

3. Table 1 : Why was surgery done for neovascularization elsewhere?

Kindly check and rephrase  these two sentences:

page 7 line 292 "RBCs deform in blood vessels to reduce flow resistance, although increased blood viscosity decreases the deformability of RBCs"

Page 8 line 315: "Unexpectedly, the group without HT in the present study did not require treatment (p = 0.04). " HTN was more in no-treatment group. Also, a mere high prevalence of HTN in N-group cannot be interpreted as HTN group requiring no-treatment.

The authors need to go through the entire manuscript and correct the grammatical errors.

Author Response

Thank you very much for your favorable reviews of our submission and helpful comments and suggestions to improve the paper. Based on our knowledge and the limited data currently at hand, we have incorporated the suggested changes in the manuscript and hope that they address the points raised sufficiently.

To Reviewer #2

  1. The number of eyes in the "T- group" is too small to provide any significant power to the results. Did the authors calculate the sample size? What is the power of the study?

 - As you pointed out, the sample size in this study was small and no statistical sample size calculations were conducted. However, for Hct a sample size of 42 patients gave post hoc powers of 0.69 to detect 4.37% of the smallest difference in the mean between the two groups under the conditions of a two-sided significance level (α) of 0.05. To test post hoc powers, JMP 13 (SAS Institute Inc., Cary, NC, USA) was used.

We also examined the post hoc powers of other parameters for which the results of the univariate analysis were p < 0.1. We confirmed that the power of detection was 0.6 or more for many parameters. However, as you pointed out, a power of 0.6 is not a sufficient value, and therefore we added the low power as a limitation of this study.

  1. Methods: a. Page 2 line 71: The word "invisible" needs to be rephrased. The indications of PPV need to be explained in detail. Surgery is not required for all types of fibrovascular membranes. The authors need to specify.

- As you suggested, we deleted the word “invisible” and added the following information on the indication for PPV:

“Indications for PPV included the following: recurring vitreous haemorrhage (VH), despite the use of maximal panretinal photocoagulation (PRP); dense premacular subhyaloid haemorrhage; combined tractional and rhegmatogenous retinal detachment by the fibrovascular membrane (FVM); and tractional retinal detachment involving or threatening the macula.” (page 5, para 1, lines 4–8).

  1. Methods: b. How were the criteria for diagnosis of macular edema? This needs to be explained further. Was OCT done? was it diagnosed clinically? Were the macular thickness measurements done? Were they centre-involving or non-centre involving?

- As you pointed out, we found insufficient information for DME diagnosis. We added the following additional information (page 7, para 1, lines 6–11):

“In all eyes, postoperative central subfoveal thickness (CST) was derived from the software (Cirrus 3.0; Carl Zeiss Meditec, Inc., Dublin, CA, USA.) provided by the manufacturer. Postoperative ME was defined as more than 300 μm CST occurring within 12 months after surgery.”

  1. The multivariate logistic regression tables are unclear. What are the dependent variables in these two tables?

- Thank you for the suggestion. As you noted, the dependent variables were not mentioned. The dependent variables in these two tables are the presence of anti-VEGF drug vitreous and/or STTA injection within 12 months after surgery.

  1. Table 1: Why was surgery done for neovascularization elsewhere?

- As you pointed out, the presence of NVE alone is not an indication for PPV. This case had recurring vitreous hemorrhage despite PRP and therefore PPV was performed in this case.

  1. Kindly check and rephrase these two sentences:

page 7 line 292 "RBCs deform in blood vessels to reduce flow resistance, although increased blood viscosity decreases the deformability of RBCs"

-We agree that this phrase might be hard to understand and therefore it was rephrased as:

“RBC deformation in blood vessels reduces flow resistance, although increased blood viscosity decreases the deformability of RBCs.” (page14, 2nd para, line9 to page15, 1st para, line1)

Page 8 line 315: "Unexpectedly, the group without HT in the present study did not require treatment (p = 0.04). " HTN was more in no-treatment group. Also, a mere high prevalence of HTN in N-group cannot be interpreted as HTN group requiring no-treatment.

-Thank you. As you pointed out, this sentence does not make sense. We have reduced the number of cases in response to reviewer 1's suggestion and analyzed the data carefully. Therefore, we found no significant difference in hypertension (P = 0.09) and a significant difference in diastolic blood pressure (P=0.03). We have revised the text to reflect the changes as following:

“Contrary to expectations, the group with high diastolic blood pressure in the present study did not require treatment (p = 0.03). High diastolic blood pressure is a risk factor for DR, heart disease, and CVD.35, 36 It was also reported to be a risk factor for DME.36, 37” (page16, 1st para, line1-3)

  1. The authors need to go through the entire manuscript and correct the grammatical errors.

- Thank you very much. We reviewed the manuscript carefully, and a native English-speaking proofreader also went through the manuscript.

Round 2

Reviewer 2 Report

The queries have been well addressed. There are no further comments.